



# Parametrization of stochastic multiscale triads

Jeroen Wouters[1,2], Stamen I. Dolaptchiev[3], Valerio Lucarini[2,4,5], and Ulrich Achatz[3]

[1]School of Mathematics and Statistics, The University of Sydney, Sydney, Australia
[2]Klimacampus, Meteorologisches Institut, University of Hamburg, Hamburg, Germany
[3]Institut für Atmosphäre und Umwelt, Goethe-Universität Frankfurt, Frankfurt am Main, Germany
[4]Department of Mathematics and Statistics, University of Reading, Reading, UK
[5]Walker Institute for Climate System Research, University of Reading, Reading, UK

*Correspondence to:* Jeroen Wouters (jeroen.wouters@uni-hamburg.de)

**Abstract.** We discuss applications of a recently developed method for model reduction based on linear response theory of weakly coupled dynamical systems. We apply the weak coupling method to simple stochastic differential equations with slow and fast degrees of freedom. The weak coupling model reduction method results in general in a non-Markovian system, we therefore discuss the Markovianization of the system to allow for straightforward numerical integration. We compare the

5 applied method to the equations obtained through homogenization in the limit of large time scale separation between slow and fast degrees of freedom. We numerically compare the ensemble spread from a fixed initial condition, correlation functions and exit times from a domain. The weak coupling method gives more accurate results in all test cases, albeit with a higher numerical cost.

## 1 Introduction

Many models of physical systems are too complex to be solved analytically, or even numerically if a large range of temporal and spatial scales is involved. For some high-dimensional dynamical systems it is however possible to derive lower-dimensional reduced models (Givon et al., 2004; Huisinga et al., 2003). The reduced model is easier to solve analytically and faster to integrate numerically, while still preserving some of the essential characteristics of the full system. This line of research lies at the heart of many applications, for example in molecular dynamics (Hijón et al., 2009; Lu and Vanden-Eijnden, 2014) and

climate modeling (Lucarini et al., 2014; Imkeller and Von Storch, 2001; Palmer and Williams, 2009).

The derivation of a reduced model is possible, for example, in the presence of a time scale separation between slow resolved and fast unresolved variables, as is assumed in the homogenization method (Pavliotis and Stuart, 2008). This method applies to slow-fast systems of the form

$$
\begin{aligned}
\dot{x} &= f_0(x,y) + \frac{1}{\varepsilon} f_1(x,y) \\
\quad \dot{y} &= \frac{1}{\varepsilon^2} g_1(x,y) + \frac{1}{\varepsilon} \beta(y)\xi(t),
\end{aligned}
\tag{1}
$$

in the limit of infinite time scale separation $\varepsilon \to 0$, where $\xi$ denotes a standard Brownian motion (i.e. the equations should be considered equivalent to a stochastic integral in the Itô interpretation) (Khas'minskii, 1963; Papanicolaou, 1976). It is evident



from the dynamical equation that the $y$ variables evolve on a faster time scale than the $x$ variables. For finite values of $\varepsilon$ there is an intricate feedback between the evolution of the $x$ and $y$ variables. The situation simplifies in the limit of $\varepsilon \to 0$ where the slow variables do not evolve on the time scales on which $y$ strongly fluctuates. As a result, the slow dynamics converges to a stochastic evolution, where the effect of $y$ is completely replaced by statistical quantities related to the motion

of $y$ for a fixed value of $x$. On a more technical note, the precise expression for the quantities entering in the reduced dynamics can be easily obtained through an expansion in $\varepsilon$ of the backward Kolmogorov equation (the adjoint of the Fokker-Planck equation) $\partial_t v(x,t) = (\mathcal{L}_0 + \mathcal{L}_1/\varepsilon + \mathcal{L}_2/\varepsilon^2)v(x,t)$ of corresponding to the slow-fast dynamics (where $\mathcal{L}_0 = f_0\partial_x$, $\mathcal{L}_1 = f_1\partial_x$ and $\mathcal{L}_2 = g_1\partial_y + (\beta/2)\partial_y^2$) (Pavliotis and Stuart, 2008).

The method of homogenization has found a great number of applications in different fields of physics and mathematics

(Pavliotis and Stuart, 2008). Many physical systems, however, do not feature a time scale separation. As an example, the climate system has variability on many different temporal (and spatial) scales, but no clear spectral gaps can be identified. This creates fundamental difficulties in the theoretical investigation of climate dynamics and in the construction of climate models. As a result, approximate equations are used for dealing with scales of motions belonging to a range of scales of interest, and numerical models are able to resolve explicitly only a fractions of the full range of scales. The dynamics taking place on scales

that are too small and/or fast to be resolved need to be parametrized. Consider the case of convective motion in the Earth's atmosphere. Convective clouds are significant for the climate, yet can only be resolved at a spatial resolution of 10–100 m (Sakradzija et al., 2015), whereas climate models only resolve scales of the order of 100 km (Intergovernmental Panel on Climate Change, 2013). Unresolved convective motion however features a so-called "gray zone", a range of time scales overlapping with the dynamical time scales of the resolved large scale flow (Sakradzija et al., 2015), therefore homogenization can

not be applied. It is a formidable challenge to derive dimension reduction methods that do not require a time scale separation. One should underline that when facing a lack of time scale separation, we would like to be able to construct self-adaptive parametrizations as opposed to empirical ones, so that when the resolution of a numerical model is changed we do not need to redo the exercise of fitting a reduced model.

Going beyond the familiar setting of infinite time scale separation requires a novel approach to the derivation of closed

equation for the reduced system. Recently, we have developed a model reduction technique that does not rely on the presence of such a separation (Lucarini et al., 2014; Wouters and Lucarini, 2012; Wouters and Lucarini, 2013). The alternative method for model reduction makes use of a weak coupling approach, in which response theory (Ruelle, 2009, 1997) is used to derive a closure. The systems of interest follow a dynamics determined by

$$\dot{x} = \varepsilon\psi_x(x,y) + f_x(x)$$

$$\dot{y} = \varepsilon\psi_y(x,y) + g_y(y), \tag{2}$$

where $x$ is the variable of interest. Exploiting the weak coupling form of this equation, response theory can be employed to expand expectation values of $x$-dependent observable under the invariant measure in orders of $\varepsilon$. This expansion yields a series in terms of $\varepsilon$, reminiscent of the Dyson series in scattering theory, each representing a sequence of interactions between the $x$ and $y$ subsystems, corresponding to a certain Feynman diagram.



The truncation of this series up to a given order yields an approximation of the response of the $x$ subsystem to the coupling to the $y$ subsystem. More importantly, it allows to determine the statistical quantities of the $y$ system that dictate this response. The first order correction to the dynamics of the $x$ system can be written as the expectation value $\varepsilon \int \mathrm{d}y \psi_x(x,y)\rho_y(y)$, where $\rho_y$ is the invariant density of the uncoupled $\dot{y} = g_y(y)$ dynamics. At second order two correction terms appear, one due to double $\psi_x$ interactions from $y$ to $x$, determined by a correlation function of the uncoupled $y$ dynamics, and a feedback term, determined by a response function of the uncoupled $y$ dynamics. This knowledge can then be exploited to derive a surrogate dynamics for $x$ that reproduces the effect of the coupling of $x$ to $y$ up to second order in $\varepsilon$. While this theory has been originally developed assuming that the uncoupled systems are Axiom A dynamical systems, it can be equally applied in the case where the uncoupled dynamics is stochastic, the only needed requirement being to have a physical measure. Interestingly, the results obtained using response theory match what one can derive by constructing a perturbative expansion of the dynamics of the system using the Mori-Zwanzig projection method (Wouters and Lucarini, 2013).

Previously, we have proposed a surrogate dynamical equation for the $x$ variable that introduces an $\varepsilon$-dependent perturbing term to the dynamics $f_x$ to match the response of the statistics of the full system. The perturbing term contains a non-Markovian memory term and a correlated noise, with the memory kernel and correlation functions depending on the statistics of the uncoupled dynamics $\dot{y} = g_y$. In a recent study of the applicability of the weak coupling approach to a simple ocean-atmosphere system, the method has been shown to give a good result for sufficiently weak coupling between the ocean and the atmosphere (Demaeyer and Vannitsem, 2016), even if it is clear that a systematic investigation of the performance of the weak coupling approach is indeed still needed.

We remark that Chekroun et al. (Chekroun et al., 2015b, a) have recently proved that, indeed, constructing reduced order models entails introducing deterministic, stochastic and memory correction to the dynamics of the variables of interest.

Here we will apply and extend the weak coupling approach of (Wouters and Lucarini, 2012; Wouters and Lucarini, 2013) for the development of parameterizations for various stochastic triad models. Triad interactions arise from quadratic nonlinearities with energy conserving properties (see e.g., (Gluhovsky and Tong, 1999)). The triad models considered here appear in applications of the homogenization technique to construction of parameterizations in climate modeling (see e.g., (Majda et al., 2001, 2002; Franzke et al., 2005; Franzke and Majda, 2006; Achatz et al., 2013; Dolaptchiev et al., 2012)). The non-Markovian memory kernel in the weak coupling approach will be calculated for these simple stochastic multiscale models and approximated by a Markovian stochastic process, in order to allow for easier numerical implementation. The systems we investigate can be written in both the weak coupling form of Eq. 2 and the slow-fast form of Eq. 1, therefore direct comparison is possible and will be performed on a number of metrics, namely initial ensemble spread, correlation functions and exit times from an interval.

## 2   The additive triad

The first model we look at is the stochastically forced additive triad. This system is a low-dimensional model that has non-linear interactions reminiscent of those occurring between the Fourier modes of a fluid flow. It is stochastically forced to mimic the



interaction with further unresolved modes. The system has three variables, one slow variable $x$ and two fast variables $y_1$ and $y_2$. The fast dynamics is dominated by two independent Ornstein-Uhlenbeck processes. The dynamical equations for this triad are

$$
\begin{aligned}
\frac{dx}{dt} &= B^{(0)} y_1 y_2 \\
\frac{dy_1}{dt} &= B^{(1)} x y_2 - \frac{\gamma_1}{\varepsilon} y_1 + \frac{\sigma_1}{\sqrt{\varepsilon}} \xi_1(t) \\
\frac{dy_2}{dt} &= B^{(2)} x y_1 - \frac{\gamma_2}{\varepsilon} y_2 + \frac{\sigma_2}{\sqrt{\varepsilon}} \xi_2(t).
\end{aligned}
\tag{3}
$$

The processes $\xi_i$ are independent Brownian motions in the Itô sense. Here and below a differential equation featuring a Brownian motion will be interpreted as the equivalent stochastic integral. In addition, we require $\sum_i B^{(i)} = 0$, which guarantees energy conservation in the case $\gamma_i = \sigma_i = 0$.

## 2.1 Homogenization

On the time scale $t$, when increasing the time scale separation $1/\varepsilon$ to infinity, we have trivial dynamics of the averaged equations $\dot{x} = B^{(0)} \langle y_1 y_2 \rangle_{\rho_{\text{OU}}} = 0$ where $\rho_{\text{OU}}$ is the Gaussian invariant measure of the fast Ornstein-Uhlenbeck process generated by taking $B^{(i)} = 0$ for $i = 1, 2, 3$. In the setting of homogenization, one looks at the convergence of the distribution of paths on a longer time scale. The time is scaled to the diffusive time scale $\theta = \varepsilon t$ and on this longer diffusive time scales deviations from the averaged dynamics develop.

By expanding the backward Kolmogorov equation for the slow-fast system in orders of $\varepsilon$, a Kolmogorov equation for only the slow variables can be derived (see (Pavliotis and Stuart, 2008)). The dynamical equation corresponding to this Kolmogorov equation is in this case a one-dimensional Ornstein-Uhlenbeck process (Majda et al., 2002)

$$
\frac{\partial x}{\partial \theta} = C_m x + \sqrt{2A_0} \xi(\theta),
\tag{4}
$$

where

$$
\begin{aligned}
C_m &= \frac{B^{(0)}}{\gamma_1 + \gamma_2} \left( B^{(1)} \frac{\sigma_2^2}{2\gamma_2} + B^{(2)} \frac{\sigma_1^2}{2\gamma_1} \right) \\
A_0 &= \frac{B^{(0)2}}{\gamma_1 + \gamma_2} \frac{\sigma_1^2}{2\gamma_1} \frac{\sigma_2^2}{2\gamma_2}.
\end{aligned}
$$

See Fig. 1 for an illustration of the homogenization principle for the additive triad. The mean and variance of the triad converge to those of the Ornstein-Uhlenbeck process (4) for small $\varepsilon$.

## 2.2 Weak coupling limit

We will now discuss the weak coupling method as described in (Wouters and Lucarini, 2012; Wouters and Lucarini, 2013). By rescaling the time as $\tau = \varepsilon^{-1} t$ we can write the stochastically forced additive triad equation (3) as a two-dimensional





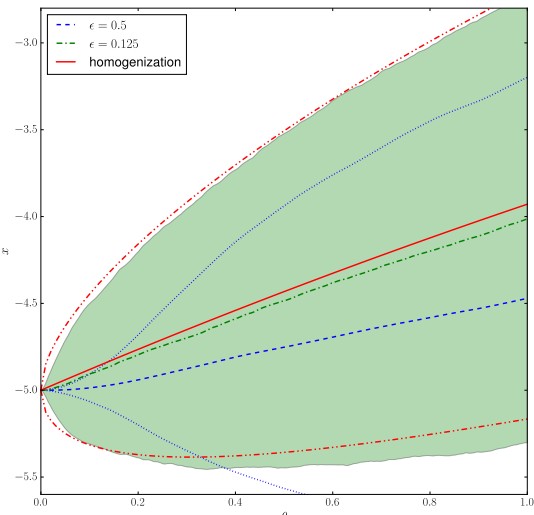

**Figure 1.** Convergence to the homogenized equations for the additive triad (3) in $\theta = \varepsilon t$ time scale. The red solid and dash-double-dotted lines show the mean and $2\sigma$ intervals respectively for an ensemble evolving according to the homogenized equation (4) from an initial condition $x = -5$. The blue dashed and dotted lines show the mean and $2\sigma$ intervals for an ensemble of the additive triad (3) for $\varepsilon = 0.5$ from an initial condition $(x, y_1, y_2) = (-5, 0, 0)$ with $B^{(0)} = -0.75$, $B^{(1)} = -0.25$, $B^{(2)} = 1$, $\gamma_1 = 1/\delta$, $\sigma_1 = \sqrt{2/\delta}$, $\gamma_2 = 1$ and $\sigma_2 = \sqrt{2}$ with $\delta = 0.75$. The green dash-dotted line and the green shaded area show the same for $\varepsilon = 0.125$.

Ornstein-Uhlenbeck system weakly coupled non-linearly to a trivial zero-gradient $x$ system:

$$
\begin{aligned}
\frac{dx}{d\tau} &= \varepsilon \psi_x(y_1, y_2) \\
\frac{dy_1}{d\tau} &= \varepsilon \psi_{y,1}(x, y) - \gamma_1 y_1 + \sigma_1 \xi_1(\tau) \\
\frac{dy_2}{d\tau} &= \varepsilon \psi_{y,2}(x, y) - \gamma_2 y_2 + \sigma_2 \xi_2(\tau).
\end{aligned}
\tag{5}
$$

5  with $\psi_x(y_1, y_2) = B^{(0)} y_1 y_2$ and $\psi_y(x, y) = (B^{(1)} xy_2, B^{(2)} xy_1)^T$. The stochastic parametrization derived in (Wouters and Lucarini, 2012; Wouters and Lucarini, 2013) is given by a non-Markovian equation

$$
\frac{d\tilde{x}}{d\tau} = \varepsilon \sigma(\tau) + \varepsilon^2 \int_0^\infty ds R(s, \tilde{x}(\tau - s)),
\tag{6}
$$





where the the memory kernel $R(s, \tilde{x})$ and first two moments of the stochastic process $\sigma(\tau)$ are derived using the weak coupling method to the following statistics of the uncoupled $y$ Ornstein-Uhlenbeck dynamics:

$$\langle \sigma(\tau) \rangle = 0$$

$$C(\tau) := \langle \sigma(0)\sigma(\tau) \rangle = \langle \psi_x(y_1, y_2)\psi_x(y_1(\tau), y_2(\tau)) \rangle_{\rho_{OU}} \tag{7}$$

$$R(\tau) = \langle \psi_y(x, y_1, y_2).\nabla_y \psi_x(y_1(\tau), y_2(\tau)) \rangle_{\rho_{OU}}. \tag{8}$$

where the evolution of $y_1$ and $y_2$ into $y_1(\tau)$ and $y_2(\tau)$ are taken to be the uncoupled Ornstein-Uhlenbeck dynamics $dy_i/d\tau = -\gamma_i y_i + \sigma_i \xi_i$. We have for the case of the additive triad (3)

$$C(\tau) = (B^{(0)})^2 \langle y_1(0)y_1(\tau) \rangle \langle y_2(0)y_2(\tau) \rangle_{\rho_{OU}} = (B^{(0)})^2 \exp(-(\gamma_1 + \gamma_2)\tau)\frac{\sigma_1^2}{2\gamma_1}\frac{\sigma_2^2}{2\gamma_2} \tag{9}$$

and

$$
\begin{aligned}
R(\tau, x) =& B^{(0)} B^{(1)} x \langle y_2(0)(\partial_{y_1} y_1(\tau))y_2(\tau) \rangle_{\rho_{OU}} \\
&+ B^{(0)} B^{(2)} x \langle y_1(0)y_1(\tau)(\partial_{y_2} y_2(\tau)) \rangle_{\rho_{OU}} \\
=& x B^{(0)} \exp(-(\gamma_1 + \gamma_2)\tau)\left(\frac{\sigma_2^2}{2\gamma_2}B^{(1)} + \frac{\sigma_1^2}{2\gamma_1}B^{(2)}\right).
\end{aligned} \tag{10}
$$

### 2.2.1 Markovian parametrization

Due to the identical time-scale $\gamma_1 + \gamma_2$ in both memory and noise correlation, the memory equation (6) can be transformed to a Markovian parametrization. We want to find a parametrizing two level Markovian dynamical system of the form

$$
\begin{aligned}
\frac{dz_1}{d\tau} &= \varepsilon C_1 z_2 \\
\frac{dz_2}{d\tau} &= -\gamma z_2 + \sigma_z \xi(\tau) + \varepsilon C_2 z_1.
\end{aligned} \tag{11}
$$

such that the second order response of this system to changes in $\varepsilon$ is the same as the response of (6). In other words, we want to determine the parameters $C_1$, $C_2$, $\gamma$ and $\sigma_z$ in (11) such that the correlation and memory functions of the fast equation in (11) are equal to (9) and (10) respectively. The correlation function $C(\tau)$ and memory function $R(\tau)$ of the fast equation of (11) are

$$C(\tau) = \langle (C_1 z_2(0))(C_1 z_2(\tau)) \rangle = C_1^2 e^{-\gamma\tau}\frac{\sigma_z^2}{2\gamma} \tag{12}$$

$$R(\tau, z_1) = \langle (C_2 z_1)\partial_{z_2}(C_1 z_2(\tau)) \rangle = C_1 C_2 z_1 e^{-\gamma\tau}, \tag{13}$$



where the evolution of $z_2$ to $z_2(\tau)$ is now given by $dz_2/d\tau = -\gamma z_2 + \sigma_z \xi(\tau)$. By equating these functions to their counterparts in (9) and (10) we see that by choosing

$$
\begin{aligned}
C_1 &= B^{(0)} \\
C_2 &= \frac{\sigma_2^2}{2\gamma_2} B^{(1)} + \frac{\sigma_1^2}{2\gamma_1} B^{(2)} = \beta_2 B^{(1)} + \beta_1 B^{(2)} \\
\gamma &= \gamma_1 + \gamma_2 \\
\sigma_z^2 &= 2 \frac{\sigma_1^2}{2\gamma_1} \frac{\sigma_2^2}{2\gamma_2} (\gamma_1 + \gamma_2) = 2\beta_1 \beta_2 \gamma
\end{aligned}
$$

the reduced $z_1$ dynamics of the parametrized dynamical system in the weak coupling method are of the same form as those of the stochastic triad (3).

This Markovian reduced equation (11) is in fact a reformulation of the non-Markovian equation (6). To see this, we write an explicit solution for $z_2$ in function of the history of $z_1$ and $\xi$ as

$$
z_2(\tau) = e^{-\gamma\tau} z_2(0) + \int_0^\tau dt' (\sigma_z \xi(t') + \varepsilon C_2 z_1(t')) e^{-\gamma(\tau - t')}.
$$

This solution can then be inserted into (11), to obtain

$$
\frac{dz_1}{d\tau} = \varepsilon C_1 e^{-\gamma\tau} z_2(0) + \varepsilon C_1 \int_0^\tau dt' (\sigma_z \xi(t') + \varepsilon C_2 z_1(t')) e^{-\gamma(\tau - t')}, \tag{14}
$$

which agrees with (6), the first two terms being an Ornstein-Uhlenbeck process with the required correlation plus a memory term with the required memory kernel.

This Markovian formulation allows for a straightforward numerical implementation of the parametrization, compared to the non-Markovian equation (6) which requires one to store the history of the process in memory.

A comparison of the performance of the two model reductions is show in Figure 2. Shown are the spread of an ensemble initiated at a fixed value for the slow variables $x = z_1 = -5$ and the autocorrelation function of the slow variables. The weak coupling method clearly gives better results.

By correctly rescaling time and taking the limit of $\varepsilon \to 0$ in the Markovian parametrization (11) one can furthermore verify that in this limit it converges to the homogenization of the original triad equation (Eq. (4)).

## 3   The slowly oscillating additive triad

The additive triad as specified in Eq. (3) can be generalized to allow for an additional interaction between the $y$ variables on the slow time scale that is independent of $x$. The dynamical equations for this slowly oscillating triad are

$$
\begin{aligned}
\frac{dx}{dt} &= B^{(0)} y_1 y_2 \\
\frac{dy_1}{dt} &= B^{(1)} y_2 x - \frac{\gamma_1}{\varepsilon} y_1 + \omega y_2 + \frac{\sigma_1}{\sqrt{\varepsilon}} \xi_1(t) \\
\frac{dy_2}{dt} &= B^{(2)} x y_1 - \frac{\gamma_2}{\varepsilon} y_2 - \omega y_1 + \frac{\sigma_2}{\sqrt{\varepsilon}} \xi_2(t).
\end{aligned} \tag{15}
$$



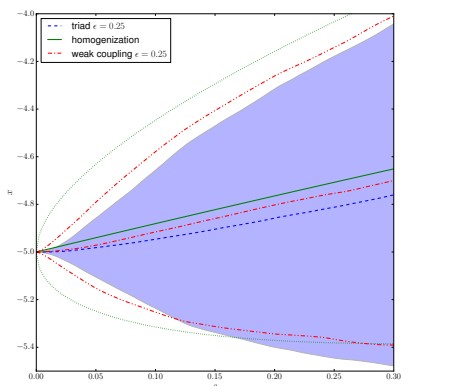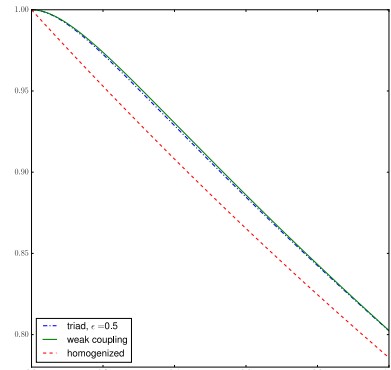

**Figure 2.** Left: comparison of the ensemble spread for the original additive triad system for $\varepsilon = 0.25$ from an initial condition $(-5, 0, 0)$ (the ensemble mean is the blue dashed line, $2\sigma$ interval the blue shaded area), the two-level Ornstein-Uhlenbeck process from the weak coupling method (11) from an initial condition $(-5, 0)$ (ensemble mean: red dash-dotted line, $2\sigma$ interval: red dash-dot-dotted lines) and the one-level Ornstein-Uhlenbeck process from homogenization (4) from $x = -5$ (ensemble mean: green solid line, $2\sigma$ interval: dotted lines)

Right: comparison of the autocorrelation functions of the slow variable $\langle x(t)x(0) \rangle$ in the full triad for $\varepsilon = 0.5$ (blue dash-dotted line), $\langle z_1(t)z_1(0) \rangle$ in the weak coupling model (green solid line) and $\langle x(t)x(0) \rangle$ for the homogenized equation (red dashed line).

Both plots use parameter values $B^{(0)} = -0.75$, $B^{(1)} = -0.25$, $B^{(2)} = 1$, $\gamma_1 = 1/\delta$, $\sigma_1 = \sqrt{2/\delta}$, $\gamma_2 = 1$ and $\sigma_2 = \sqrt{2}$ with $\delta = 0.75$.

## 3.1 Homogenization

The homogenized equation is similar to the one for the additive triad, with an added constant forcing $C_r$ in the reduced SDE

$$\frac{\partial x}{\partial \theta} = C_m x + C_r + \sqrt{2A_0}\xi(t)$$

$$C_r = \frac{B^{(0)}}{\gamma_1 + \gamma_2}\omega\left(\frac{\sigma_2^2}{2\gamma_2} - \frac{\sigma_1^2}{2\gamma_1}\right).$$

5 ## 3.2 Weak coupling limit

The coupling functions $\psi_x$ and $\psi_y$ are now

$$\psi_x(y) = B^{(0)}y_1 y_2$$

$$\psi_y(x, y) = x\begin{pmatrix} B^{(1)}y_2 \\ B^{(2)}y_1 \end{pmatrix} + \omega\begin{pmatrix} y_2 \\ -y_1 \end{pmatrix}.$$

The correlation function (7) of the coupling to $x$, determining the correlations of the parametrization noise $\sigma$ is

$$\langle \psi_x(y)\psi_x(f^\tau(y)) \rangle = B^{(0)^2}\langle y_1 f^\tau(y_1) \rangle \langle y_2 f^\tau(y_2) \rangle$$

$$= B^{(0)^2}\exp(-(\gamma_1 + \gamma_2)s)\frac{\sigma_1^2}{2\gamma_1}\frac{\sigma_2^2}{2\gamma_2}.$$





The response function (8) of $\psi_x$ to $\psi_y$, determining the memory kernel of the parametrization, is similar to the one for the additive triad, with an added exponential function, the integral of which gives the same constant $C_r$ of the homogenized equations

$$
\begin{aligned}
R(\tau, x) &= \langle \psi_y(x,y) \partial_y \psi_x(y(\tau)) \rangle \\
&= \exp(-\gamma\tau)(D_1 x + D_0) \\
D_1 &= B^{(0)} \left( B^{(1)} \frac{\sigma_2^2}{2\gamma_2} + B^{(2)} \frac{\sigma_1^2}{2\gamma_1} \right) = \gamma C_m \\
D_0 &= \omega B^{(0)} \left( \frac{\sigma_2^2}{2\gamma_2} - \frac{\sigma_1^2}{2\gamma_1} \right) = \gamma C_r.
\end{aligned}
$$

Combined, this then results in the following non-Markovian parametrized equations

$$
\begin{aligned}
\frac{d\tilde{x}}{d\tau} &= \varepsilon\sigma(\tau) + \varepsilon^2 \int_0^\infty \mathrm{d}s\, R(s, \tilde{x}(\tau-s)) \\
&= \varepsilon\sigma(\tau) + \varepsilon^2 \int_0^\infty \mathrm{d}s\, \exp(-\gamma s)(D_1 \tilde{x}(\tau-s) + D_0) \\
&= \varepsilon\sigma(\tau) + \varepsilon^2 \int_0^\infty \mathrm{d}s\, \exp(-\gamma s)\tilde{x}(\tau-s) + \varepsilon^2 C_r.
\end{aligned}
\tag{16}
$$

### 3.2.1 Markovian parametrization

The non-Markovian equation (16) can again be Markovianized by a two-level Ornstein-Uhlenbeck process of the form

$$
\begin{aligned}
\frac{dz_1}{d\tau} &= \varepsilon C_1 z_2 \\
\frac{dz_2}{d\tau} &= -\gamma z_2 + \sigma_z \xi(t) + \varepsilon(C_2 z_1 + C_3).
\end{aligned}
\tag{17}
$$

The corresponding correlation and memory terms are

$$
C(\tau) = C_1^2 e^{-\gamma\tau} \frac{\sigma_z^2}{2\gamma}
\tag{18}
$$

$$
R(\tau) = C_1 e^{-\gamma\tau}(C_2 z_1 + C_3).
\tag{19}
$$

We can therefore take

$$
\begin{aligned}
C_3 &= D_0/C_1 \\
&= \omega \left( \frac{\sigma_2^2}{2\gamma_2} - \frac{\sigma_1^2}{2\gamma_1} \right).
\end{aligned}
$$

In the limit $\varepsilon \to 0$ in the Markovian parametrization (17) we again recover the homogenized equations.

### 3.3 Exit times

When comparing initial ensemble spread and autocorrelation functions for the slow variable of this system with the weak coupling parametrization and the homogenized system, the results are similar to those presented for the additive triad above.





Additionally, here we perform a comparison of a rare event statistic, the first exit time of the slow variable from an interval $[-1, 1]$ when the slow variable is initialized at 0.

| $\varepsilon$ | 0.5 | 0.25 | 0.125 |
|---|---|---|---|
| homogenization | 0.403 | 0.184 | 0.0982 |
| weak coupling | 0.205 | 0.0839 | 0.0589 |

**Table 1.** The relative error on the mean exit time $|\mathbb{E}_1(\tau) - \mathbb{E}_0(\tau)|/\mathbb{E}_0(\tau)$ where $\mathbb{E}_0(\tau)$ is the mean exit time from $[-1, 1]$ of the full triad system and $\mathbb{E}_1(\tau)$ is the mean exit time of the parametrized systems with $B^{(0)} = -0.75$, $B^{(1)} = -0.25$, $B^{(2)} = 1$, $\omega = 0.25$, $\gamma_1 = 1/\delta$, $\sigma_1 = \sqrt{2/\delta}$, $\gamma_2 = 1$ and $\sigma_2 = \sqrt{2}$ with $\delta = 0.75$.

| $\varepsilon$ | 0.5 | 0.25 | 0.125 |
|---|---|---|---|
| homogenization | 0.420 | 0.217 | 0.115 |
| weak coupling | 0.232 | 0.0814 | 0.0395 |

**Table 2.** The relative error on the standard deviation of the exit times $|\sigma_1(\tau) - \sigma_0(\tau)|/\sigma_0(\tau)$ where $\sigma_0(\tau)$ is the standard deviation of exit times from $[-1, 1]$ of the full triad system and $\sigma_1(\tau)$ is the standard deviation of exit times of the parametrized systems. Parameters are chosen as in Table 1.

The results in Tables 1 and 2 show that the statistics of exit times are significantly better approximated in the weak coupling parametrization.

## 4   The rapidly oscillating additive triad

A further generalization of the additive triad (3) is to introduce an interaction between the $y$ variables on the fast time scale (Dolaptchiev et al., 2012). The dynamical equations for the rapidly oscillating triad are

$$
\begin{aligned}
\frac{dx}{dt} &= B^{(0)} y_1 y_2 \\
\frac{dy_1}{dt} &= B^{(1)} y_2 x - \frac{\gamma_1}{\varepsilon} y_1 + \frac{\omega}{\varepsilon} y_2 + \frac{\sigma_1}{\sqrt{\varepsilon}} \xi_1(t) \\
\frac{dy_2}{dt} &= B^{(2)} x y_1 - \frac{\gamma_2}{\varepsilon} y_2 - \frac{\omega}{\varepsilon} y_1 + \frac{\sigma_2}{\sqrt{\varepsilon}} \xi_2(t).
\end{aligned}
\tag{20}
$$

Note the difference in scaling on the oscillatory terms $\omega y_i$ compared to Eq. (15). The invariant measure of the fast system is a correlated Gaussian measure $\rho(y) = \exp(-y^T (2R)^{-1} y)/\mathcal{Z}$ determined by

$$
\Gamma R + (\Gamma R)^T = \Sigma^T \Sigma
$$

with

$$
\Gamma = \begin{pmatrix} \gamma_1 & -\omega \\ \omega & \gamma_2 \end{pmatrix}
$$



and

$$\Sigma = \begin{pmatrix} \sigma_1 & 0 \\ 0 & \sigma_2 \end{pmatrix}.$$

Homogenization leads to a solvability condition on the system 20 that is fulfilled if either $\omega = 0$ or $\sigma_1^2/\gamma_1 = \sigma_2^2/\gamma_2$. The homogenized equation is now given by

$$\dot{x} = C_\omega x + \sqrt{2A_\omega}\xi(t)$$

with

$$
\begin{aligned}
C_\omega &= bB^{(1)}R_{22} + 2(aB^{(1)} + cB^{(2)})R_{12} + bB^{(2)}R_{11} \\
A_\omega &= B^{(0)}(3aR_{11}R_{12} + b(R_{11}R_{22} + R_{12}^2) + 3cR_{22}R_{12}) \\
b &= \frac{B^{(0)}}{\left(\frac{\omega^2}{\gamma_1} + \frac{\omega^2}{\gamma_2} + \gamma_1 + \gamma_2\right)} \\
a &= (-\omega/2\gamma_1)b \\
c &= (\omega/2\gamma_2)b.
\end{aligned}
$$

### 4.1 Weak coupling

The coupling functions of Eq. (20) have the following form

$$\psi_x(y_1, y_2) = B^{(0)}y_1 y_2 \tag{21}$$

$$\psi_y(x, y_1, y_2) = x(B^{(1)}y_2, B^{(2)}y_1)^T. \tag{22}$$

The correlation function $\langle \psi_x(y_1, y_2)\psi_x(y_1(t), y_2(t))\rangle$ appearing in the weak coupling expansion can again be calculated explicitly. Solutions of the fast Ornstein-Uhlenbeck system $\dot{y} = -\Gamma y + \Sigma \xi$ can be written as

$$y_i(t) = [e^{-\Gamma t}y(0)]_i + \int_0^t d\tau [e^{-\Gamma(t-\tau)}\Sigma \dot{W}(\tau)]_i.$$

Inserting this expression into the autocorrelation function gives

$$
\begin{aligned}
\sigma_\omega(t) := \langle \psi_x(y_1, y_2)\psi_x(y_1(t), y_2(t))\rangle &= (B^{(0)})^2 \langle y_1(0)y_2(0)y_1(t)y_2(t)\rangle \\
&= (B^{(0)})^2 \Big([e^{-\Gamma t}]_{11}[e^{-\Gamma t}]_{21}(3R_{11}R_{12}) \\
&\quad + \left([e^{-\Gamma t}]_{11}[e^{-\Gamma t}]_{22} + [e^{-\Gamma t}]_{12}[e^{-\Gamma t}]_{21}\right)(R_{11}R_{22} + 2R_{12}^2) \\
&\quad + [e^{-\Gamma t}]_{12}[e^{-\Gamma t}]_{22}(3R_{22}R_{12})\Big) \\
&\quad + (B^{(0)})^2 R_{12} \int_0^t d\tau_1 d\tau_2 \langle [e^{-\Gamma t}\Sigma\xi(\tau_1)]_1[e^{-\Gamma t}\Sigma\xi(\tau_2)]_2\rangle,
\end{aligned}
$$




since the noise $\xi$ is white and has zero mean.

The memory term $h$ can be calculated by performing integration by parts on the response function, resulting in a fluctuation-dissipation type expression:

$$
\begin{aligned}
h_\omega(\tau) &= \left\langle \left( -\frac{\nabla.(\rho\psi_y)}{\rho} \right) \psi_x(\tau) \right\rangle \\
&= B^{(0)}x \left\langle \left( B^{(1)}[R^{-1}]_{12}y_2^2(0) + (B^{(1)}[R^{-1}]_{11} + B^{(2)}[R^{-1}]_{22})y_1(0)y_2(0) + B^{(2)}[R^{-1}]_{12}y_1^2(0) \right) y_1(\tau)y_2(\tau) \right\rangle
\end{aligned}
$$

### 4.1.1 Markovian parametrization

Guided by the Markovian form of the previous triad systems, we again want to derive a Markovian parametrization with a reduced one-level Ornstein-Uhlenbeck system as the fast component:

$$
\begin{aligned}
\dot{z}_1 &= \varepsilon C_1 z_2 \\
\dot{z}_2 &= \varepsilon C_2 z_1 - \gamma z_2 + \sigma_z \xi_z(t).
\end{aligned}
\tag{23}
$$

In this case, there is no exact match between the auto-correlation and response functions of this Markovian system and the non-Markovian weak coupling parametrization. The choice of the parametrization parameters is therefore not exactly determined and one needs to choose a parametrization such that the auto-correlation and response functions of the coupling function in the fast component of the full system are approximated in some sense. A further restriction comes from the fact that in the limit $\varepsilon \to 0$ the limiting path distribution of the full system is determined by the homogenized equation and we therefore want to retain this limiting behavior in the parametrized system. To have this limiting property, we have the following constraints on the parameters in Eq. (23)

$$
\begin{aligned}
\frac{C_1^2\sigma_z^2}{2\gamma^2} &= A_\omega \\
\frac{C_1 C_2}{\gamma} &= C_\omega,
\end{aligned}
$$

where $A_\omega$ and $C_\omega$ are the forcing and friction parameters obtained through homogenization. For formal equivalence between the reduced and full equations, we furthermore set $C_1 = B^{(0)}$. With the remaining free parameters we can match the response and correlation functions in a more precise manner, for example by matching the values of these functions at time $t = 0$. In this way, we get

$$
C_2 = \frac{h_y(0)}{B^{(0)}}
$$

and

$$
\sigma_z^2 = \frac{2\gamma_{\mathrm{wc}}\sigma_\omega(0)}{B^{(0)2}},
$$

where $h_y = h_\omega/x$.

A simulation of the ensemble spread from a fixed initial condition is shown in Figure 3. It demonstrates that the weak coupling parametrization (23) outperforms the homogenized reduced system.





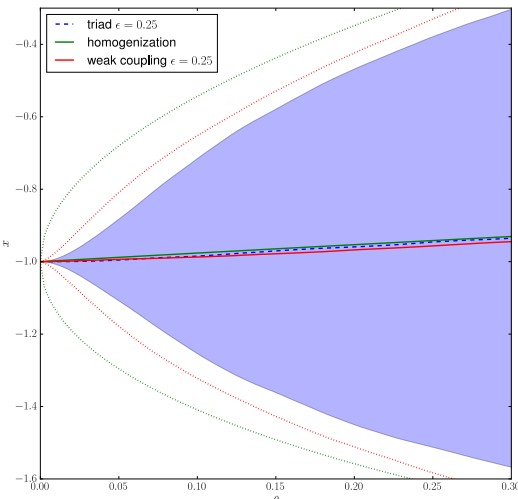

**Figure 3.** comparison of the ensemble spread for the original oscillating triad system for $\varepsilon = 0.25$ from an initial condition (-5,0,0) (the ensemble mean is the blue dashed line, $2\sigma$ interval the blue shaded area), the two-level Ornstein-Uhlenbeck process from the weak coupling method (11) from an initial condition (-5,0) (ensemble mean: red dash-dotted line, $2\sigma$ interval: red dash-dot-dotted lines) and the one-level Ornstein-Uhlenbeck process from homogenization (23) from $x = -5$ (ensemble mean: green solid line, $2\sigma$ interval: dotted lines) $B^{(0)} = -0.75$, $B^{(1)} = -0.25$, $B^{(2)} = 1$, $\omega = 1/12$, $\gamma_1 = 1/\delta$, $\sigma_1 = \sqrt{2/\delta}$, $\gamma_2 = 1$ and $\sigma_2 = \sqrt{2}$ with $\delta = 0.75$.

## 4.2 Exit times

The same experiment on exits from an interval has been performed as described in Section 3.3. The results are displayed in Table 3. As before, the weak coupling reduced system gives a much better result when compared to the homogenized system.

| $\varepsilon$ | 0.5 | 0.25 | 0.125 |
|---|---|---|---|
| homogenization | 0.534 | 0.262 | 0.118 |
| weak coupling | 0.322 | 0.127 | 0.0619 |

**Table 3.** The relative error on the mean exit time $|\mathbb{E}_1(\tau) - \mathbb{E}_0(\tau)|/\mathbb{E}_0(\tau)$ where $\mathbb{E}_0(\tau)$ is the mean exit time from $[-1, 1]$ of the full triad system and $\mathbb{E}_1(\tau)$ is the mean exit time of the parametrized systems. The parameters are the same as those used for Fig. 3.



| $\varepsilon$ | 0.5 | 0.25 | 0.125 |
|---|---|---|---|
| homogenization | 0.583 | 0.286 | 0.118 |
| weak coupling | 0.362 | 0.109 | 0.0503 |

**Table 4.** The relative error on the standard deviation of the exit times $|\sigma_1(\tau) - \sigma_0(\tau)|/\sigma_0(\tau)$ where $\sigma_0(\tau)$ is the standard deviation of exit times from $[-1, 1]$ of the full triad system and $\sigma_1(\tau)$ is the standard deviation of exit times of the parametrized systems. The parameters are the same as those used for Fig. 3.

## 5 The multiplicative triad

A final type of interactions is given by the multiplicative triad equations (Majda et al., 2002)

$$
\begin{aligned}
\frac{dx_1}{dt} &= B^{(1)} x_2 y \\
\frac{dx_2}{dt} &= B^{(2)} x_1 y \\
\frac{dy}{dt} &= B^{(3)} x_1 x_2 - \frac{\gamma_m}{\varepsilon} y + \frac{\sigma_m}{\sqrt{\varepsilon}} \xi(t),
\end{aligned}
\tag{24}
$$

which describes the interplay between two $x$ modes and a stochastically forced single $y$ mode. In the absence of forcing and dissipation energy conservation is satisfied if $\sum_i B^{(i)} = 0$. In the system (24) the $y$ mode can be eliminated directly by integrating the last equation of (24)

$$
y(t) = e^{-\frac{\gamma_m}{\varepsilon} t} y(0) + \int_0^t dt' \left( \frac{\sigma_m}{\sqrt{\varepsilon}} \xi(t') + B^{(3)} x_1(t') x_2(t') \right) e^{-\frac{\gamma_m}{\varepsilon}(t-t')}.
$$

Inserting this result in the equations for the $x$ variables, one obtains

$$
\frac{d}{dt} \begin{pmatrix} x_1(t) \\ x_2(t) \end{pmatrix} = \begin{pmatrix} B^{(1)} x_2(t) \\ B^{(2)} x_1(t) \end{pmatrix} \left\{ e^{-\frac{\gamma_m}{\varepsilon} t} y(0) + \int_0^t dt' \left( \frac{\sigma_m}{\sqrt{\varepsilon}} \xi(t-t') + B^{(3)} x_1(t-t') x_2(t-t') \right) e^{-\frac{\gamma_m}{\varepsilon} t'} \right\}.
\tag{25}
$$

Note that the first two term on the righthand side. result from a Ornstein-Uhlenbeck process with zero mean and stationary time autocorrelation function given by $\frac{\sigma_m^2}{2\gamma_m} e^{-\frac{\gamma_m}{\varepsilon} t}$.

### 5.1 Weak coupling

The coupling functions for the multiplicative triad read

$$
\begin{aligned}
\psi_x(x,y) &= (B^{(1)} x_2 y, B^{(2)} x_1 y)^T, \\
\psi_y(x) &= B^{(3)} x_1 x_2.
\end{aligned}
$$

The coupling terms in the $x$ equations are separable

$$
\psi_{x,i}(x,y) = a_i \psi'_{x,1,i}(x) \psi'_{x,2,i}(y)
\tag{26}
$$



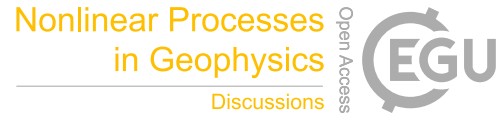

with $\langle \psi'_{x,2,i}(y) \rangle_{\rho_{OU}} = 0$, where

$$a_1 = B^{(1)}, \quad \psi'_{x,1,1}(x) = x_2, \quad \psi'_{x,2,1}(y) = y,$$
$$a_2 = B^{(2)}, \quad \psi'_{x,1,2}(x) = x_1, \quad \psi'_{x,2,2}(y) = y.$$

The resulting parametrization in the weak coupling approach (Wouters and Lucarini, 2012; Wouters and Lucarini, 2013) reads

$$5 \quad \frac{dx_i}{d\tau} = \varepsilon a_i \psi'_{x,1,i} \sigma_i(\tau) + \varepsilon^2 \int_0^\infty \mathrm{d}s R_i(s, x(\tau - s)), \tag{27}$$

with a noise term $\sigma_i$ with zero mean and correlation given by

$$\langle \sigma_i(0) \sigma_j(\tau) \rangle = \langle \psi'_{x,2,i}(y) \psi'_{x,2,j}(y(\tau)) \rangle_{\rho_{OU}} = \frac{\sigma_m^2}{2\gamma_m} e^{-\gamma_m \tau}.$$

The memory kernel has the form

$$R_i(s,x) = \langle \psi_y(x,y) \cdot \nabla_y \psi_{x,i}(x(s), y(s)) \rangle_{\rho_{OU}},$$

$$10 \qquad R(s,x) = B^{(3)} x_1 x_2 e^{-\gamma s} \begin{pmatrix} B^{(1)} x_2(s) \\ B^{(2)} x_1(s) \end{pmatrix}$$

Thus (27) can be written as

$$\frac{d}{d\tau} \begin{pmatrix} x_1(\tau) \\ x_2(\tau) \end{pmatrix} = \begin{pmatrix} B^{(1)} x_2(\tau) \\ B^{(2)} x_1(\tau) \end{pmatrix} \left\{ \sigma(\tau) + \int_0^\infty \mathrm{d}s B^{(3)} x_1(\tau - s) x_2(\tau - s) e^{-\gamma_m s} \right\}, \tag{28}$$

which is exactly the same result as in (25), if we rescale time and assume as initial condition $x_1(t) = x_2(t) = 0$ for $t < 0$. In this case the weak coupling approach recovers exactly the full model. The original three component system was reduced to a two

15 component non-Markovian system but there is no efficiency gain using the parametrization since the corresponding Markovian system is again a three component one.

## 5.2 Homogenization

Introducing a longer time scale $\theta = \varepsilon^2 \tau$ in (28) and taking the limit $\varepsilon \to 0$ one recovers the homogenization result in Stratonivich formulation

$$20 \quad \frac{d}{d\theta} \begin{pmatrix} x_1 \\ x_2 \end{pmatrix} = \frac{B^{(3)}}{\gamma} x_1 x_2 \begin{pmatrix} B^{(1)} x_2 \\ B^{(2)} x_1 \end{pmatrix} + \frac{\sigma_m}{\gamma_m} \begin{pmatrix} B^{(1)} x_2 \\ B^{(2)} x_1 \end{pmatrix} \xi(\theta). \tag{29}$$

The latter corresponds to an Itô stochastic differential equation of the form

$$\frac{d}{d\theta} \begin{pmatrix} x_1 \\ x_2 \end{pmatrix} = \frac{B^{(3)}}{\gamma} x_1 x_2 \begin{pmatrix} B^{(1)} x_2 \\ B^{(2)} x_1 \end{pmatrix} + \frac{\sigma_m^2}{2\gamma_m^2} B^{(1)} B^{(2)} \begin{pmatrix} x_1 \\ x_2 \end{pmatrix} + \frac{\sigma_m}{\gamma_m} \begin{pmatrix} B^{(1)} x_2 \\ B^{(2)} x_1 \end{pmatrix} \xi(\theta). \tag{30}$$

For a comparison of the statistics of the multiplicative triad and of the homogenization model we refer to (Majda et al., 2002).



## 6 Conclusions

In this work we have worked out a first application of the weak coupling response method of (Wouters and Lucarini, 2012; Wouters and Lucarini, 2013) to a multiscale stochastic system. By the choice of system we were able to perform both homogenization and the weak coupling reduction on this system, thereby allowing for direct comparison between the two reductions.

The response method yields a non-Markovian equation, making it cumbersome to integrate numerically. We have demonstrated here that for the systems studied the non-Markovian equation can be further reduced to a Markovian equation. Even with this further reduction the system gives a better match to the original system than the homogenized equations.

In the case of the additive triads, the system that is obtained through the Markovianization procedure is of intermediate complexity, between the full system and the homogenized limit. In the systems studied here, the retention of a fast time scale

in the reduced system means that the reduction in simulation complexity is modest (one variable instead of two and a linear coupling instead of a nonlinear one). In the case of the multiplicative triad the weak coupling parametrization recovers exactly the full model and there is no efficiency gain. In many applications of practical relevance, however, one considers situations where the number of degrees of freedom of the unresolved variables is considerably larger than those of the slow variables of interest. A reduction to a system of one or a few variables will constitute a significant reduction in complexity in this case. This

approach can be compared to the superparametrization approach to convection, where convection is parametrized by a model that is still dynamical in nature, yet significantly simpler than the full convective motion (Randall et al., 2003; Grooms and Majda, 2013, 2014).

*Acknowledgements.* JW is grateful to Georg Gottwald and Cesare Nardini for stimulating discussions. VL is grateful to M Chekroun, C Franzke, and M Ghil for a lot of food for thought on the problem of constructing reduced model in geophysical fluid dynamical systems.

The research leading to these results has received funding from the European Community's Seventh Framework Programme (FP7/2007-2013) under grant agreement n° PIOF-GA-2013-626210, as well as from the DFG project MERCI. SD if thankful to the German Research Foundation (DFG) for partial support through DO 1819/1-1. UA thanks the German Research Foundation (DFG) for partial support through grant AC 71/7-1.



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
