# Peer review of "Parametrization of stochastic multiscale triads"

_Nonlinear Processes in Geophysics, 2016_

## Referee Comment (RC1) · Anonymous Referee #1 · 27 Jul 2016

I have reviewed the manuscript entitled "Parametrization of stochastic multiscale triads" (npg-2016-37). I enjoyed this paper and thought the exploration of the weak coupling approach as compared to homogenization was prudent, especially if the goal is to use this tool in future stochastic averaging problems. The paper is laid out in a straightforward way with a strong emphasis that it is showcasing the application of a particular method (rather than discussion of the method itself) and the appropriate references are provided regarding the method. Overall, I think this paper does a good job of demonstrating the applicability of the weak coupling method and its virtues compared to homogenization. The metrics of comparison are appropriate and I think constitute sufficient evidence for the conclusions the authors have drawn (i.e. that for simple stochastic triad models, weak coupling is a more consistent reduction method than homogenization). I also appreciate that the authors discuss the limitations of the weak

coupling method for the multiplicative triad, giving this paper a proper grounding. Based on the manuscript and the research scope of the journal, I believe this work falls within the desired scope of NPG and meets the review criteria.

I have a few minor suggestions that I think would make this work more complete and informative to the reader. They are mostly intended for those approaching this topic for the first time. I have listed most of them below in a point form.

My advance thanks to the authors for considering my comments. Their responses to my comments are appreciated. My compliments to the authors for their work.

Scientific / Content –

1. p2, line 11 - There is at least one work out there that shows the lack of spectral gap explicitly in climate dynamics. Readers may appreciate a reference to this fact. The work of J. M. Mitchell (1976), although it is not new, offers a discussion of this.

2. p2, line 33 - Is there a good reference for scattering theory that be offered here?

3. p2, line 34 - This is the famous Feynman diagram from particle physics, correct? If so, no change is necessary. I just wanted to confirm.

4. p3, line 8 - What are the characteristics of Axiom A dynamical system that make the theory of Dyson series appropriate? (I assume it is the chaotic nature of Axiom A systems, but maybe a sentence here would be helpful)

5. p3, line 9 - Can you clarify what is meant by a physical measure? My immediate intuition leads me to think this means either a delta-function measure or a finitely integrable measure. I assume the author means the latter, as a delta-function measure does not generally apply to stochastic systems.

6. p8, line 10 - Does $f^{\tau}$ have a particular signifcance? Or is it just a place holder function? It appears to be a time-evolution operator to me. If so, then what is the variable 's' on the following line? This may require some clarification.

7. p11, line 3 - Can the authors comment on the solvability condition? i.e. Is the condition that the mean of the system drift is equal to zero?

8. p12, line 20 - For formal equivalence, you are comparing the coefficients of the homogenized model and Markovian weak coupling model, correct? I'm not sure that "formal equivalence" between the full and reduced equations is the most clear way to explain this. Would it be more appropriate to describe the equivalence as "statistical" or "weak" (as opposed to "strong").

9. I would be interested in any comments the authors can make about the asymptotic nature of the weak behaviour of the reduced systems compared to the full systems in terms of the long time behaviour. (i.e. For Figure 3, what happens as theta goes to infinity? Does the superiority of the weak coupling model continue to hold?) The comments made through the additive examples suggest that for epsilon small enough the weak coupling model converges to the homogenization model but the lines in Figure 3 give me the impression that for large epsilon, the weak coupling model may do better initially, but the homogenization may give a better long time approximation in terms of distribution width.

10. I have a concern about the scaling of the fast-slow system (1) compared to system (3). The powers of epsilon are not equivalent between the two systems and I suspect that this has implications for the scalings in systems (2) and (5) which are equivalent in epsilon. Should the reader be concerned about the difference in scalings?

11. For completeness, I recommend that the authors include the ensemble sizes used to generate the curves in the figures.

12. Regarding Tables 1-4, a mean and standard deviation are probably sufficient to demonstrate the superiority of the weak coupling method for the systems considered, but I think it would be interesting to see the distribution of exit times in terms of a PDF. If this is easily done, I would be curious to see it.

Technical –

1. Graphs were challenging to read on paper because the axis labels and lines were small and/or thin. If the authors could make the plots more readable, I'm sure readers would appreciate it.

2. Equations are not consistently labeled. (i.e. 20 / (20) / Eq. 20). I also notice that the authors sometime make reference to the full triad systems without equation refs, where as the reduced systems often have equation refs. For increased readability, I think it would be good to include equation references to the full systems where possible.

3. p6, line 20 - Should R(\tau) be R(\tau,z)? There are a few other instances where the memory function, R, is specified with either one or two arguments. It would be good to keep this consistent.

4. p15, line 5 - The variables sigma_{1}, sigma_{2} have already been used on page 4 as scaling coefficients for the noise terms. I think a different variable would help the clarity here.

5. p5, line 6 - Regarding the previous point, I also think it would be appropriate to change the variable sigma(\tau) to something else.

---

## Referee Comment (RC2) · Anonymous Referee #2 · 24 Aug 2016

This work compares the performance of two methods to derive the reduced dynamical equations of a more complex dynamical system, by applying these methods to ODEs which describe stochastically forced additive and multiplicative triads. The first method is the homogenization method, which is designed to work for systems with a pronounced time-scale separation between the (slow) variable of interest **x** and fast variables **y** whose detailed evolution need not be computed. The second method recently developed by the authors is the so-called weak coupling method that derives the approximate response of **x** subsystem to the **y** subsystem, and does not rely on the time-scale separation between the two subsystems, which corresponds much better to geophysical reality. The weak coupling approach results in models that outperform those derived using homogenization approach, at the expense of relative numerical efficiency.

[Figure]

This is a very nice, important work, and the paper is succinct and well written. I recommend publication and I am looking forward to the authors' applying their methodology to intermediate geophysical models.

Minor comments:

p. 1, title: Shouldn't this be "parameterization" ("e" after "t")?

p. 2, l. 7: ...of corresponding to... –> remove 'of'

p. 2, ll. 21-23: there is nothing wrong with empirical parameterizations either!

p.3, ll. 24-25: remove extra parentheses

p. 15, l. 23: I think this comparison should at least be summarized in a sentence or two, then the reader should be referred to Majda's reference for further details

p. 16, l. 4: 'two reductions' –> two reduction methods

p. 16, btw ll. 17 and 18: Any words on work in progress/future work? Can this be applied to a system of intermediate complexity, with many simultaneous triad interactions?

---

## Author Comment (AC1) · 20 Oct 2016

Significant changes to the text have been marked in blue in the attached pdf.

**1   Referee 1**

**1.1   Scientific / Content**

1. *p2, line 11 - There is at least one work out there that shows the lack of spectral gap explicitly in climate dynamics. Readers may appreciate a reference to this fact. The work of J. M. Mitchell (1976), although it is not new, offers a discussion*

*of this.*

Thank you for pointing out this reference.

It has been added to the text (p.2 line 12).

2. *p2, line 33 - Is there a good reference for scattering theory that be offered here?*

   We have added a reference to an introductory account of scattering theory and Feynman diagrams (p.2 line 34).

3. *p2, line 34 - This is the famous Feynman diagram from particle physics, correct? If so, no change is necessary. I just wanted to confirm.*

   Yes, that is correct.

4. *p3, line 8 - What are the characteristics of Axiom A dynamical system that make the theory of Dyson series appropriate? (I assume it is the chaotic nature of Axiom A systems, but maybe a sentence here would be helpful)*

   The existence of linear and higher order response has been rigorously proven for Axiom A by Ruelle. This allows us to make use of the perturbational approach.

   A remark on this has been added to the text (p.3 line 8).

5. *p3, line 9 - Can you clarify what is meant by a physical measure? My immediate intuition leads me to think this means either a delta-function measure or a finitely integrable measure. I assume the author means the latter, as a delta-function measure does not generally apply to stochastic systems.*

   A clarification of the term physical measure has been added to the text (p.3 line 11).

6. *p8, line 10 - Does $f^\tau$ have a particular signifcance? Or is it just a place holder function? It appears to be a time-evolution operator to me. If so, then what is the variable 's' on the following line? This may require some clarification.*

The variable 's' was a typo, this should have been $\tau$. The time-evolution operator has been remove to unify the notation with the rest of the text.

7. *p11, line 3 - Can the authors comment on the solvability condition? i.e. Is the condition that the mean of the system drift is equal to zero?*

   Yes, this is correct.

   A comment has been added to the text (p.11 line 8)

8. *p12, line 20 - For formal equivalence, you are comparing the coefficients of the homogenized model and Markovian weak coupling model, correct? I'm not sure that "formal equivalence" between the full and reduced equations is the most clear way to explain this. Would it be more appropriate to describe the equivalence as "statistical" or "weak" (as opposed to "strong").*

   The weak coupling method does not give sufficient constraints to fully determine the coefficients of the Markovian model. However, from additional numerical experiments, the results presented in the article appear to be stable under a change of the coefficient $C_1$. An initial calculation based on the Mori-Zwanzig method suggests the same, although this is a matter for further research.

   The text has been changed to reflect this (p.13 line 6).

9. *I would be interested in any comments the authors can make about the asymptotic nature of the weak behaviour of the reduced systems compared to the full systems in terms of the long time behaviour. (i.e. For Figure 3, what happens as theta goes to infinity? Does the superiority of the weak coupling model continue to hold?) The comments made through the additive examples suggest that for epsilon small enough the weak coupling model converges to the homogenization model but the lines in Figure 3 give me the impression that for large epsilon, the weak coupling model may do better initially, but the homogenization may give a better long time approximation in terms of distribution width.*

A test for the rapidly oscillating triad for much longer integration times (35 time units) shows no significant difference between the two methods beyond 2 time units. It is interesting though that even though the ensemble spread beyond this point is similar for the homogenization method and the weak coupling method, there is still an improvement in exit time distribution up to 30 time units.

A remark has been added to the text (p.13 line 11).

10. *I have a concern about the scaling of the fast-slow system (1) compared to system (3). The powers of epsilon are not equivalent between the two systems and I suspect that this has implications for the scalings in systems (2) and (5) which are equivalent in epsilon. Should the reader be concerned about the difference in scalings?*

The system (3) is in fact of the form (1), by a scaling of time.

A remark on this has been added to the text (p.4 line 10).

11. *For completeness, I recommend that the authors include the ensemble sizes used to generate the curves in the figures.*

This has been added in the figure captions.

12. *Regarding Tables 1-4, a mean and standard deviation are probably sufficient to demonstrate the superiority of the weak coupling method for the systems considered, but I think it would be interesting to see the distribution of exit times in terms of a PDF. If this is easily done, I would be curious to see it.*

A plot of the cumulative histogram of exit times has been inserted in section 4 (top of p. 14).

**1.2 Technical**

1. *Graphs were challenging to read on paper because the axis labels and lines were small and/or thin. If the authors could make the plots more readable, I'm sure readers would appreciate it.*

   Plots with thicker lines and larger fonts have been substituted.

2. *Equations are not consistently labeled. (i.e. 20 / (20) / Eq. 20). I also notice that the authors sometime make reference to the full triad systems without equation refs, where as the reduced systems often have equation refs. For increased readability, I think it would be good to include equation references to the full systems where possible.*

   The references to equations has been unified. We have also added references to the equations in question in many place throughout the text.

3. *p6, line 20 - Should R(\tau) be R(\tau,z)? There are a few other instances where the memory function, R, is specified with either one or two arguments. It would be good to keep this consistent.*

   These should all be corrected now.

4. *p15, line 5 - The variables sigma_{1}, sigma_{2} have already been used on page 4 as scaling coefficients for the noise terms. I think a different variable would help the clarity here.*

   The variable $\sigma$ has been changed to $\eta$.

5. *p5, line 6 - Regarding the previous point, I also think it would be appropriate to change the variable sigma(\tau) to something else.*

   The variable $\sigma$ has been changed to $\eta$.
**2 Referee 2**

1. *p. 1, title: Shouldn't this be "parameterization" ("e" after "t")?*

   Both ways of spelling are valid according to various dictionaries and are used in the scientific literature.

2. *p. 2, l. 7: . . .of corresponding to. . . –> remove 'of'*

   This has been removed.

3. *p. 2, ll. 21-23: there is nothing wrong with empirical parameterizations either!*

   We don't claim anything to be 'wrong' with empirical parametrization.

4. *p.3, ll. 24-25: remove extra parentheses*

   This has been removed.

5. *p. 15, l. 23: I think this comparison should at least be summarized in a sentence or two, then the reader should be referred to Majda's reference for further details*

   Additional details on the results have been added to the text (p.16 line 11).

6. *p. 16, l. 4: 'two reductions' –> two reduction methods*

   Corrected.

7. *p. 16, btw ll. 17 and 18: Any words on work in progress/future work? Can this be applied to a system of intermediate complexity, with many simultaneous triad interactions?*

   This is indeed work in progress, which will be reported on when ready.

Please also note the supplement to this comment:
http://www.nonlin-processes-geophys-discuss.net/npg-2016-37/npg-2016-37-AC1-supplement.pdf

[Figure]

**Supplement:**

[revised manuscript text omitted]